# Direct–Maternal Genetic Parameters for Litter Size and Body Weight of Piglets of a New Black Breed for the Taiwan Black Hog Market

**DOI:** 10.3390/ani12233295

**Published:** 2022-11-25

**Authors:** Hsiu-Lan Lee, Mu-Yao Lin, Han-Sheng Wang, Chin-Bin Hsu, Cheng-Yung Lin, Shen-Chang Chang, Perng-Chih Shen, Hsiu-Luan Chang

**Affiliations:** 1Animal Industry Division, Livestock Research Institute, Council of Agriculture, Executive Yuan, Tainan 71246, Taiwan; 2Graduate Institute of Bioresources, National Ping Tung University of Science and Technology, Pingtung 91201, Taiwan; 3Formosan Farmers Association for Swine Improvement, Taipei 24242, Taiwan; 4Taitung Animal Propagation Station, Livestock Research Institute, Council of Agriculture, Executive Yuan, Taitung 95444, Taiwan; 5Nutrition Division, Livestock Research Institute, Council of Agriculture, Executive Yuan, Tainan 71246, Taiwan; 6Kaohsiung Animal Propagation Station, Livestock Research Institute, Council of Agriculture, Executive Yuan, Pingtung 91247, Taiwan; 7Department of Animal Science, National Pingtung University of Science and Technology, Pingtung 91201, Taiwan

**Keywords:** animal model, genetic parameter, piglet performance, maternal genetic effects, sow

## Abstract

**Simple Summary:**

Black pigs are the most competitive hogs in Taiwan as they have a higher price than commercial non-black hogs due to their desirable meat quality for locals, and they have accounted for 15% of Taiwan’s hog production since 2011. However, the litter sizes before the weaning of local black sows are much smaller than those of maternal breeds such as Landrace and Yorkshire, which are commonly used in Taiwan. Herd productivity is highly influenced by dam litter size and piglet weight before weaning. The KHAPS Black Pig was registered as a new breed in Taiwan in 2010 and was designed as a maternal line for black hog production. The direct–maternal genetic effects on litter size and piglet weight were tested for selection efficiency. The direct–maternal genetic correlation had a great impact on the prediction of maternal breeding value. Therefore, it is crucial to estimate the direct–maternal genetic parameters of litter size and piglet weight to assist future selection decisions for the genetic progress of this new breed.

**Abstract:**

The objective of this study was to estimate the genetic parameters of litter size and piglet weight from farrowing to weaning in KHAPS Black sows. The genetic parameters investigated were the direct (h^2^_d_), maternal (h^2^_m_), realized (h^2^_r_), and total (h^2^_T_) heritability, as well as correlations (r_d_, r_m_, and r_dm_) within and between traits. The analyses were performed using single- and three-trait animal models with and without maternal genetic effects. In the three-trait model with maternal genetic effect, all estimates of h^2^_d_ and h^2^_m_ were significantly different from zero except the h^2^_d_ of mean birth weight. Positive values of r_d_ and r_m_ between traits were observed as expected in the range of 0.322–1.000. Negative values of r_dm_ were found within and between traits and were less associated with mean piglet weight traits than litter size traits. Estimates of h^2^_T_ were consistently larger than those of h^2^_r_ in both the single- and three-trait model analyses. In addition, the three-trait model can take into account the association between the traits, so the estimates are more accurate with smaller SEs. In conclusion, maternal genetic effects were not negligible in this study, and thus, a multiple-trait animal model with maternal genetic effects and full pedigree is recommended to assist future pig breeding decisions in this new breed.

## 1. Introduction

Black pigs are the most competitive hogs in Taiwan due to their high meat quality, and they have accounted for 15% of Taiwan’s hog production since 2011. The KHAPS Black Pig breed was developed via Meishan–Duroc crossbreeding with a Meishan maternal lineage in a breeding program focused on sow litter performance that was conducted at the Kaohsiung Animal Propagation Station of Taiwan’s Livestock Research Institute. The KHAPS Black Pig population was created after a 12-year (1997–2009) developmental research program and was admitted to registration by the Council of Agriculture, Taiwan in 2010 and then released to the public as a maternal line for local black hog production. Although the carcass characteristics of this new black lop-eared breed are more desirable than those of Duroc, Hampshire, and other local black breeds, its littering performance still has room for improvement when compared to those of maternal breeds in Taiwan such as Landrace and Yorkshire.

Sow productivity is the foundation of commercial pork production. Piglets weaned per sow per year (PSY) is one of key factors for evaluating sow productivity and thus the efficiency of pig farming. The littering performance of sows, including litter size and piglet weight at weaning, are features that have a significant impact on the fertility, prolificacy, and reproductive longevity of females. Generally, maternal effects occur when the phenotype is influenced by its dam’s phenotype in addition to the genes it inherits, and thus they have a particular impact on the offspring when it comes to maternal-influenced traits, e.g., litter size and weight from birth until weaned [1]. Ignoring maternal effects in a genetic evaluation model may result in biased estimates of genetic parameters, such as the inflated direct heritability estimates reported for growth traits of beef cattle by Meyer [2] and upward biased estimates of weaning weight [3,4]. However, significant and negligible maternal genetic effects on litter size in pigs have been reported in the literature for genetic and non-genetic reasons, such as different populations, different environmental conditions, and with or without cross-fostering. Roehe and Kennedy [5] reported non-negligible maternal effects with a negative correlation with direct genetic effects using a simulation study, which was consistent with results observed in first-parity litters of Landrace and Yorkshire breeds along with a considerable response to litter size selection by Southwood and Kennedy [6]. Contradictory results were also found by Mercer and Crump [7] that showed non-significant maternal genetic effects on litter size in British Landrace pigs. Perez-Enciso and Gianola [8] considered parity a fixed effect in their evaluation model and did not find significant maternal genetic effects on litter size in Iberian pig strains.

In addition to maternal genetic variance, covariance between direct genetic effects and maternal genetic effects may also exist with a magnitude that varies by species [1,3,5]. In the presence of maternal effects, the total breeding value (TBV) of an individual is the sum of the direct and maternal breeding value, which includes its own breeding value as the maternal effect, not that of its dam [9]. The reason for this is that the breeding animal transmits its own (not its dam’s) breeding value for maternal effects to its offspring, and thus the TBV represents the value of breeding an animal for genetic improvement so that the heritable impact and the corresponding total heritable variance (*σ*^2^*_TBV_*) can generate a selection response. The objective of this study was to investigate the importance of maternal effects and to evaluate the effect of the direct–maternal genetic parameters on the maternal-influenced traits of females, including litter size and weight of piglet from birth to weaned, in KHAPS Black Pigs.

## 2. Materials and Methods

All individuals were from the swine experimental farm of the Kaohsiung Animal Propagation Station, Taiwan. A total of 7388 piglets from 688 litters farrowed by 268 sows from 2013 to 2019 were recorded and analyzed based on sow level. There were no littermates found in the 268 sows due to only one gilt being selected per litter for mating, and thus the common-litter effect was not applicable. Moreover, an optimum estimate of permanent environmental variance was not obtained, and this parameter was therefore excluded from the statistical model. A total of 5381 and 5305 piglets were weighed at 3 and 4 weeks (weaned) of age, respectively. Two datasets, each with three traits, were then analyzed. The three traits were the total number of piglets born (LS0) and the number of piglets at 3 and 4 weeks (weaning) of age (LS3 and LS4). Piglet weights were pre-calculated as within-litter mean piglet weight for birth weight (MW0) and adjusted for 21 and 28 days of age before the within-litter mean piglet weight calculation for 3 and 4 weeks of age, MW3 and MW4. Statistical analysis was carried out at the female breeding stock level. Descriptive statistics were obtained using the PROC UNIVARIATE procedure in SAS [10].

Single-trait and three-trait animal models with or without maternal genetic effects were employed within the datasets. The derivative-free restricted maximum likelihood procedure described by Groeneveld et al. [11] was used to estimate the variance–covariance components and thus the phenotypic variances, heritability (including direct, maternal, realized, and total), and correlation. The pedigree was traced back more than 10 generations and included a total of 20,829 animals with 313 parents (112 were sires and 201 were dams). In the pedigree, 24.29% of the animals were inbred, with an average inbreeding coefficient of 4.56%. A total of 1.78% of the inbred animals had the highest inbreeding coefficient ranging from 25 to 30%, while 68.28% of the inbred animals had an inbreeding coefficient that was less than 5%.

Before finalizing the model, the PROC GLM feature of SAS was used to test the significance of the fixed effects (birth year and parity) and animal effects (using RANDOM statement). The results showed that a significant (α) level were <0.0001 for birth year and animals, and <0.05 for parity. Therefore, the multiple-trait linear model with maternal genetic effects was defined as the full model and formed as follows:*y* = *X_β_ β* + *Z_d_d* + *Z_m_m* + *e*

where *y*, *β*, *d*, *m*, and *e* are vectors of the observations of traits, fixed effects (birth year with 7 classes, and parity with 2 classes (primiparous vs. multiparous)), random direct genetic effects, random maternal genetic effects, and random residuals, respectively; *X_β_*, *Z_d_*, and *Z_m_* are incidence matrices relating vectors *β*, *d*, and *m* with *y*. All random effects are distributed as centered normal distributions with the variance–covariance matrices being equal to:$$V\left[\begin{array}{c} d\\ m\\ e \end{array}\right]=\left[\begin{array}{ccc} G_{d}⊗A & G_{dm}⊗A & 0\\ G_{dm}⊗A & G_{m}⊗A & 0\\ 0 & 0 & R⊗I \end{array}\right]$$
where ⊗ is the Kronecker product operator, *A* is the additive genetic relationship matrix (pedigree traced back to the foundation generation, more than 10 generations), and *G_d_*, *G_m_*, *G_dm_*, and *R* represent the (co)variance matrices between the traits for direct genetic effects, maternal genetic effects, associations between direct and maternal genetic effects, and residual effects, respectively. *I* is an identity matrix of the appropriate size. Other models, such as single-trait models with or without maternal genetic effects, were simplified from the full model above.

The variance components for direct genetic effects (*σ*^2^_d_), maternal genetic effects (*σ*^2^_m_), the covariance between direct and maternal genetic effects (*σ*_dm_), and residuals (*σ*^2^_e_) were used to calculate the phenotypic variance following Dickerson [12], Willham [13], and Eaglen and Bijma [9] as *σ*^2^_y_ = *σ*^2^_d_ + *σ*^2^_m_ +*σ*_dm_ + *σ*^2^_e_, which was based on the model of *P_i_* = *d_i_* + *m_j_* + *ε_i_* for the phenotype of individual *i* with dam *j,* where *d_i_* and *m_j_* are the direct and maternal breeding values for *i* and *j*, respectively, and *ε_i_* is the nonheritable effects of *i*. Furthermore, the total breeding value (*TBV*), according to Eaglen and Bijma [9], is the value of the individual *i* for genetic improvement, and thus *TBV_i_* = *d_i_* + *m_i_*. Therefore, the heritability was computed based on estimates of variance and covariance components as *σ*^2^_d_/*σ*^2^_y_, *σ*^2^_m_/*σ*^2^_y_, (*σ*^2^_d_ + 0.5*σ*^2^_m_ + 1.5*σ*_dm_)/*σ*^2^_y_, and (*σ*^2^_d_ + *σ*^2^_m_ + 2*σ*_dm_)/*σ*^2^_y_ for direct (h^2^_d_), maternal (h^2^_m_), realized (h^2^_r_) [12,13], and total (h^2^_T_) [9] heritability, respectively. However, some studies referred to h^2^_r_ as total heritability (h^2^_t_), although Willham [13] indicated h^2^_r_ as the fraction of the realized selection differential for mass selection.

The numerators of h^2^_r_ and h^2^_T_ represent the covariance between the *TBV*s and phenotypes of individuals and the total heritable variance, *σ*^2^*_TBV_*, respectively. Moreover, the direct and maternal genetic correlation between the *i*th and *j*th traits were estimated as r_d_ $=Cov(d_{i},d_{j})/(\sigma_{d_{i}}*\sigma_{d_{j}})$ and $r_{m}=Cov(m_{i},m_{j})/(\sigma_{m_{i}}*\sigma_{m_{j}})$. Similarly, the correlation between direct and maternal genetic effects can be obtained as $r_{dm}=Cov(d_{i},m_{j})/(\sigma_{d_{i}}*\sigma_{m_{j}})$ for within (*i* = *j*) and between (*I* ≠ *j*) trait evaluations.

## 3. Results

### 3.1. Descriptive Statistics

A descriptive summary of the littering performances evaluated is presented in Table 1. Litter sizes before weaning showed more variation than piglet weights at the corresponding age in terms of coefficient of variation (CV). CVs for litter sizes and piglet weights increased with the age of the piglets until weaning and ranged from 37 to 43% and 19 to 24%, respectively. However, neither litter size nor piglet weight until weaning substantially deviated from the expected normal distribution, as is shown in Figure 1. The skewness values ranged from −0.113 to 0.027 and from 0.38 to 0.522 for litter size traits and piglet weight traits, respectively, which were within acceptable ranges, and thus the distributions of traits studied were considered as almost symmetric distributions. Moreover, the ranges of kurtosis (actually computing excess kurtosis) values were −0.467 to −0.347 for litter sizes with a light-tailed distribution, and −0.006 to 0.424 for piglet weights with a heavy-tailed distribution except for MW0. In this study, of the 7388 piglets, 23.6% were born weighing less than 1.11 kg with around an 84% perinatal survival rate, resulting in 73 and 72% survival rates at 3 and 4 weeks of age, respectively.

### 3.2. Genetic Parameter Estimates

The heritability estimates obtained using single- and three-trait animal models without maternal genetic effects were significantly different from zero (*p* < 0.05), as is shown in the diagonals of Table 2. When maternal genetic effects were not included in the animal model, the heritability estimates obtained from three-trait model did not substantially change from those obtained when using the single-trait model. However, a slightly larger SE was shown in the single-trait model than that of the corresponding trait in the three-trait model.

Without maternal genetic effects included in the model, significant and strong positive genetic correlations between the traits were observed, which ranged from 0.790 to 0.998 (*p* < 0.001) for litter size and from 0.426 to 0.976 (*p* < 0.01) for piglet weight until weaning. Strong positive genetic correlations between females’ litter size from farrowing to weaning were found. The result might imply the existence of pleiotropy in the traits evaluated, which is consistent with the findings of Zhang et al. [14] in a case study on Meishan pigs using QTL overlapping intervals for immunity traits. However, a piglet’s survival is closely related to its immunity, which in turn affects the litter size. Moreover, Zhang et al. [14] also identified the pleiotropic genes and gene sets for growth-related traits, and thus a similar reason could be extended to correlations between pre-weaned weights, MW0, MW3, and MW4, although less strength is presented in this study. Moderate to strong phenotypic correlations were shown with ranges of 0.643–0.988 and 0.410–0.881 for litter size and piglet weight, respectively. In addition, the phenotypic associations between litter size traits were stronger than those between piglet weight, i.e., r_LS0,LS3_ = 0.797 > r_MW0,MW3_ = 0.567. The same situation is also observed in terms of the genetic correlation between traits.

In Table 3, estimates of genetic parameters from a single-trait animal model in the presence of maternal genetic effects are listed in comparison to estimates of some parameters from a three-trait model. When using single-trait analyses, much larger magnitudes of h^2^_d_ and h^2^_m_ were obtained in litter size traits than were observed in piglet weight traits except at birth. Conversely, estimates of h^2^_r_ and h^2^_T_ for piglet weight traits tended to be larger than the corresponding values of litter size traits; 0.047–0.137 vs. 0.045–0.052 for h^2^_r_ and 0.104–0.199 vs. 0.077–0.103 for h^2^_T_. A possible explanation might be the stronger negative association between direct and maternal genetic effects within litter size traits (ranged from −0.814 to −0.757) and the weaker association within piglet weight traits (ranged from −0.745 to −0.022), which might be supported by other studies [15,16]. Dong et al. [15] reported negative direct and maternal genetic correlations in Large White pigs, −0.339 and −0.668 for birth weight and litter size, respectively. Arango et al. [16] used different threshold-linear models to estimate direct and maternal genetic correlations in Large White pigs and reported intermediate and negative estimates (−0.07 to −0.31) for birth weight, but much larger estimates (−0.84 to −0.50) were found in litter size-related traits, especially for mortality during the suckling period. A similar trend for h^2^_r_ and h^2^_T_ also appeared in the results of three-trait model analysis (values shown in parentheses); 0.065–0.140 (piglet weight traits) vs. 0.038–0.066 (litter size traits) in h^2^_r_ and 0.116–0.195 (piglet weight traits) vs. 0.076–0.114 (litter size traits) in h^2^_T_.

As expected from the formulae, the direct–maternal genetic correlation could affect h^2^_r_ and h^2^_T_ estimates, and the influence depends on the direction and strength of the association (r_dm_) as well as the magnitude of the maternal genetic variance, *σ*^2^_m_. In other words, a high negative direct–maternal genetic correlation might result in low estimates of h^2^_r_ and h^2^_T_. The estimates of maternal genetic variance found in our study were larger than the absolute value of the within-trait direct–maternal genetic covariance estimate, and thus the h^2^_r_ estimated from the single- and three-trait models were 40–70% and 50–80% of h^2^_T_, respectively.

Table 4 provides genetic parameters of the sow’s litter size traits estimated using the three-trait animal model in the presence of maternal genetic effects. All estimates of genetic parameters, h^2^_d_, h^2^_m_, r_d_, r_m_, and r_dm_ for litter size traits differed from zero at the 5% significance level. Ranges of h^2^_d_ and h^2^_m_ estimates for litter size traits were similar in the univariate and multivariate animal models (0.108–0.181 vs. 0.139–0.170 for h^2^_d_ and 0.222–0.237 vs. 0.216–0.244 for h^2^_m_). They did not substantially differ between the two models. However, estimates of h^2^_m_ for litter size traits decreased with age in the univariate analysis (0.237, 0.225, and 0.222 for LS0, LS3, and LS4, respectively), but this decreasing trend was not shown in the multivariate analysis.

Estimates of r_d_ between the litter size traits were positively strong (0.924–0.999) and were slightly higher than those of r_g_ obtained from the model in the absence of maternal genetic effects (0.790–0.998 as shown in Table 2). Estimates of r_d_ and r_m_ between litter size traits were positively high, as expected, with the values being >0.85 for all traits analyzed (Table 4). The estimate of r_dm_ for within and between litter size traits were negatively strong and ranged from −0.830 to −0.696 for within traits and −0.897 to −0.801 for between traits evaluations.

As is shown in Table 5, the estimates of direct and maternal heritability for piglet weight traits were significantly different from zero except for the h^2^_d_ estimates for MW0 (0.105 ± 0.066). However, the estimates of the h^2^_d_ and h^2^_m_ for piglet weight traits in general were lower when compared to those for litter size traits except at birth weight (Table 4 and Table 5). Similar to the results of the litter size traits, estimates of direct genetic correlations between piglet weight traits were stronger than those estimated by models without maternal genetic effects (r_d_ = 0.874–0.989 in Table 5 vs. r_g_ = 0.426–0.976 in Table 2). In the multivariate analyses in the presence of maternal genetic effects, less strong maternal genetic associations were found between piglet weight traits, r_m_, compared to between litter size traits, 0.322–0.937 vs. 0.862–1.000.

The estimates of the direct–maternal genetic correlations (r_dm_) within the piglet weight traits (ranging from −0.660 to −0.084) were weaker than the estimates within the litter size traits (ranging from −0.830 to −0.696). In addition, the maternal genetic influence on piglet weight was observed to diminish with age, which appeared to be consistent with maternal genetic effects decreasing with animal age. Therefore, the direct–maternal genetic association within the trait also diminished with age; r_dm_ = −0.660, −0.431, and −0.084 at birth at 3 and 4 weeks of age, respectively.

## 4. Discussion

### 4.1. Descriptive Statistics

Since all collected records were used in the analysis and no selective recording was conducted, unbiased estimates of genetic parameters would be expected in this study. The pre-weaning survivability of piglets were 73 and 72% at 3 and 4 weeks of age, respectively, in our study, which was lower than expected. Several studies have shown that piglet birth weight is an important metric for pre-weaning survivability [17,18,19], especially during the first four days of life. Feldpausch et al. [19] further pointed out that there was a curvilinear relationship between birth weight (interval ranging from 0.5 to 2.3 kg) and pre-weaning mortality and suggested that 1.11 kg is the threshold piglet birth weight for survivability regardless of litter size. Therefore, in addition to prolificacy, birth weight should also be taken into account in the KHAPS Black Pig breeding program to promote productivity, although it was originally designed as a maternal line for black hog production in Taiwan.

### 4.2. Genetic Parameter Estimates

A three-trait analysis can take into account the correlation between traits and thus would be expected to estimate the genetic parameters, h^2^, more accurately. This is consistent with the results shown in Table 2, in which h^2^ estimated by the three-trait model has a smaller standard error than that obtained from the univariate animal model. Note that more accurate genetic parameter estimates would lead to more reliable predictions of breeding values and would consequently be beneficial to the industry’s profitability [5,6]. In this study, in the three-trait model that ignored the maternal genetic effects, the h^2^ estimates were at low to moderate levels for the traits studied with a range of 0.146–0.253 (Table 2), which is in accordance with the ranges presented in the review by Bidanel [20] and the estimated values for litter size born (h^2^_LS0_ = 0.20 ± 0.04) and piglet weight at 3 weeks of age (h^2^_MW3_ = 0.26 ± 0.06) reported by Banville et al. [21]. However, the reported h^2^ estimates of the prolificacy traits in the literature for pigs in general are about 0.10 and vary by population, as is to be expected [6,22,23,24,25].

The genetic correlation estimates (r_g_) between the traits within the model were positively favorable and high, as is shown in Table 2, so selection for any traits within the model could be expected to indirectly improve the other two traits due to the correlated selection response. Furthermore, the genetic correlation estimates between the traits were larger than the phenotypic ones; 0.790–0.998 vs. 0.643–0.988 for litter size traits and 0.426–0.976 vs. 0.410–0.881 for piglet weight traits. Environmental effects could explain these results. Under the usual additive model, phenotypic correlations (r_P_) are the sum of the genetic (G) and environmental (E) components, and causal correlations are weighted by the relative importance of heritable (*h*) and nonheritable effects (*e*). We can therefore express the r_P_ of traits X and Y as r_P_ = *h_X_ h_Y_ r_G_* + *e_X_ e_Y_ r_E_*, where *h* and *e* represent the square root of heritability (h^2^) and the square root of the proportion of phenotypic variance due to environmental factors and *r* is a correlation with the subscripts P, G, and E repsenting the phenotypic, genetic, and environmental corrleations, repectively, for traits X and Y [26]. The strong genetic correlation between litter size at birth (LS0) and at weaning (LS4), r_g_ = 0.79, combined with the weak environmental correlation between these two traits resulted in an overall phenotypic correlation that is positive but moderate, r_p_ = 0.64. Similar results were found in the case of the relationships between litter size traits and piglet weight traits presented in this study. Moreover, both the genetic and phenotypic correlations between traits decreased as the corresponding interval between the two traits recorded increased, i.e., r_LS3,LS4_ > r_LS0,LS3_ > r_LS0,LS4_, and r_MW3,MW4_ > r_MW0,MW3_ > r_MW0,MW4_. In the absence of maternal genetic effects, the estimates of the genetic correlations between the traits in this study were in agreement with reports from the literature that show a moderate to strong positive correlation [20,21,23,24,27].

In the presence of maternal genetic effects in a univariate animal model, much lower h^2^_d_ estimates were observed than those from models that ignored the maternal genetic effects. Estimates of h^2^_d_ were lower than 0.2 (0.108–0.181) and 0.1 (0.078–0.085) in terms of the litter size and piglet weight traits, respectively; these values are within the ranges of other studies [15,28,29,30]. However, a moderate h^2^_d_ estimate, 0.36, was reported by Roehe et al. [31] for birth weight in an outdoor study.

In this study, estimates of maternal heritability were on average approximately twice as large as those of direct heritability in the presence of maternal genetic effects. The smaller values of the h^2^_d_ estimate than the h^2^_m_ estimate shown in this study comport with the results reported in other studies [15,16,28,29,30,32], as well as in studies involving other species such as sheep [31]. Wilson et al. [33] found low h^2^_d_ estimates and a relatively large proportion of maternal genetic effects of birth weight and natal litter size in the Soay sheep breed, but the opposite result was also shown for birth weight and average daily gain from birth to weaning in Iranian Makooei sheep [34].

Our estimates of the h^2^_m_ for litter size traits were found to have much higher values than those reported in the literature [6,15,16,28], but they were similar to those presented by Alves et al. [30]. However, the h^2^_m_ estimates of piglet weight traits were consistent with the ranges reported in the literature, which ranged from 0.119 to 0.315 for MW0 and from 0.06 to 0.24 for MW4 [15,28,32,35].

Although the h^2^_m_ estimates from the univariate animal models were similar to the h^2^ estimates when the maternal genetic effects were ignored, the maternal influence on an individual generally decreases as an animal gets older, and thus the h^2^_m_ estimates decrease as a piglet ages; the values were 0.237 and 0.222 for LS0 and LS4 and 0.252 and 0.125 for MW0 and MW4, respectively. We should also note that higher estimates of the h^2^_d_ and h^2^_m_ were found for litter size traits than piglet weight traits in both the univariate and multivariate analyses. Similar phenomena were also observed in the strength of the direct–maternal genetic correlation (r_dm_) within and between traits when using the same analysis. The univariate analysis showed that maternal genetic effects (h^2^_m_) on litter size and piglet body weight at birth to weaning decreased as the piglets were weaned, but this was not the case for the multivariate analysis, which showed inconsistent results for litter size traits and piglet weight traits.

Negative correlation estimates (shown in Table 4 and Table 5) were observed between the direct and maternal genetic effects in all traits assessed when using both the univariate and multivariate animal models, which is in agreement with the values reported in the literature for pigs [6,34,35,36] and other species such as sheep [37] and cows [2,38]. Since litter size and piglet weight showed a negative r_dm_ both within and between traits, low h^2^_r_ and h^2^_T_ estimates would be expected. One possible explanation is that the traits are influenced by both direct and maternal genetic effects (i.e., h^2^_d_ > 0 and h^2^_m_ > 0), but these two effects oppose each other, which is consistent with others’ results [16,29,30,31,35]. Thus, the net result of this “see saw” process is that the two effects cancel each other out so that the overall h^2^_r_ and h^2^_T_ values are low or that there may even be no heritable variation.

When a comparison was made between the estimates of the h^2^_r_ and h^2^_T_, the h^2^_T_ showed a larger estimate than the h^2^_r_ due to the fact that the maternal genetic effects were greater than the direct–maternal genetic effects in this study. The h^2^_r_ and h^2^_T_ values were estimated using equations proposed by Willham [13] and Eaglen and Bijma [9], respectively. Willham [13] referred to h^2^_r_ as the fraction of the selection differential realized for phenotypic selection (h^2^_r_ = R/S = *σ*_TBV,P_/*σ*^2^_P_), which is also known as the regression coefficient of the TBV and the phenotype [12]. Eaglen and Bijma [9] defined h^2^_T_ as the fraction of an individual’s total breeding value variance (h^2^_T_ = *σ*^2^_TBV_/*σ*^2^_P_). As was stated by Eaglen and Bijma [9], in the presence of maternal genetic effects, h^2^_r_ is the proportion of phenotypic variance attributable to additive genetic effects and thus refers to the realized heritability of mass selection, whereas h^2^_T_ is the ratio of the total heritable variance (*σ*^2^_TBV_) available for a selection response to a phenotypic variation. Thus, h^2^_T_ appears to be more relevant for mammalian farm species because mass selection is rarely performed alone in livestock breeding programs.

In the presence of maternal genetic effects, the estimate of r_d_ was stronger than the corresponding estimate of the genetic correlation (r_g_) under the model that ignored the maternal genetic effects. The possible reason for this might be due to *σ*^2^_m_ and *σ*_dm_ being separated from the genetic variance with negatively strong direct–maternal genetic covariance in all traits evaluated when maternal genetic effects were considered in the model. In this study, the estimates of r_dm_ between the traits were only available in a three-trait model. Therefore, a three-trait animal model showed more advantages than a single-trait one when early or indirect selection was preferred. It was shown by the r_dm_ that weaker direct–maternal genetic associations were observed in terms of piglet body weight traits when compared with litter size traits, which implies a stronger maternal genetic influence on prolificacy than on pre-weaning growth.

## 5. Conclusions

Litter size (LS0, LS3, and LS4) and mean piglet weight (MW0, MW3, and MW4) from farrowing to weaning in KHAPS Black Pigs had direct heritability in the range of 0.11–0.18 and 0.08–0.11 and maternal heritability in the range of 0.22–0.24 and 0.10–0.25, respectively. A negative correlation between direct and maternal effects was found in all traits, but significance was found in LS0, LS3, LS4, and MW3 when using a three-trait model and in LS3 and LS4 when using a single-trait model. We conclude, therefore, that direct–maternal covariance is essential in litter size traits and mean piglet weight at birth in KHAPS Black Pigs.

## Figures and Tables

**Figure 1 animals-12-03295-f001:**
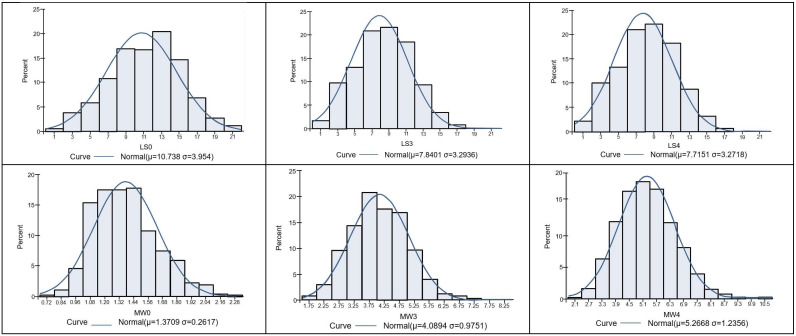
Histograms of the distribution of litter sizes (top: LS0, LS3, and LS4) and piglet weights (bottom: MW0, MW3, and MW4) from birth to 3 weeks or 4 weeks.

**Table 1 animals-12-03295-t001:** Descriptive summary for litter sizes and average piglet weights from birth to weaning in KHAPS Black Pigs.

Item	Litter size at, head			Average piglet weight at, kg		
	Birth	3-week	4-week	Birth	3-week	4-week
Minimum	1	1	1	0.74	1.81	2.24
Maximum	21	17	17	2.31	8.31	10.51
Mean	10.74	7.84	7.72	1.37	4.09	5.27
SD	3.95	3.29	3.27	0.26	0.98	1.24
CV	36.8%	42.0%	42.4%	19.0%	24.0%	23.5%
Skewness	−0.113	0.027	0.011	0.522	0.380	0.400
Kurtosis	−0.347	−0.447	−0.467	−0.006	0.424	0.278

**Table 2 animals-12-03295-t002:** Estimates of heritability (h^2^, diagonals), genetic correlations (r_g_, upper diagonals), and phenotypic correlations (r_p_, lower diagonals) of littering performance before weaning with SE using animal models without maternal genetic effects in a three-trait animal model.

Litter size at	Birth	3-week old	4-week old
Birth	0.236 ± 0.045 (0.230 ± 0.047)	0.797 ± 0.069	0.790 ± 0.072
3-week old	0.662 ± 0.029	0.253 ± 0.049 (0.250 ± 0.053)	0.998 ± 0.000
4-week old	0.643 ± 0.029	0.988 ± 0.006	0.247 ± 0.049 (0.247 ± 0.052)
Body weight at	Birth	3-week old	4-week old
Birth	0.220 ± 0.037 (0.221 ± 0.049)	0.567 ± 0.136	0.426 ± 0.144
3-week old	0.486 ± 0.033	0.148 ± 0.044 (0.146 ± 0.046)	0.976 ± 0.018
4-week old	0.410 ± 0.035	0.881 ± 0.018	0.194 ± 0.044 (0.198 ± 0.051)

Values in parentheses are estimates of heritability obtained with a single-trait model without maternal genetic effects.

**Table 3 animals-12-03295-t003:** Direct, maternal, realized, and total heritability estimates and direct–maternal genetic correlation of littering performances using a single-trait animal model with maternal genetic effects.

Trait	h^2^_d_	h^2^_m_	r_dm_	h^2^_r_	h^2^_T_
LS0	0.108 ± 0.079	0.237 * ± 0.100	−0.757 ± 0.460	0.045 (0.066)	0.103 (0.114)
LS3	0.181 * ± 0.083	0.225 ± 0.113	−0.814 * ± 0.316	0.047 (0.038)	0.077 (0.076)
LS4	0.174 * ± 0.073	0.222 ± 0.134	−0.791 * ± 0.340	0.052 (0.040)	0.085 (0.079)
MW0	0.080 ± 0.084	0.252 * ± 0.085	−0.745 ± 0.593	0.047 (0.065)	0.121 (0.125)
MW3	0.085 ± 0.079	0.115 ± 0.090	−0.484 ± 0.551	0.071 (0.089)	0.104 (0.116)
MW4	0.078 ± 0.069	0.125 ± 0.069	−0.022 ± 0.675	0.137 (0.140)	0.199 (0.195)

Values in parentheses are estimates of h^2^_r_ and h^2^_T_ obtained from a three-trait model with maternal genetic effects. * Significant at *p* < 0.05 by Z-test.

**Table 4 animals-12-03295-t004:** Direct and maternal heritability estimates (diagonals), direct–maternal genetic correlations (upper diagonals), and SE (lower diagonals) of sow’s litter size using a three-trait animal model with maternal genetic effects.

Item	Direct Genetic Effects			Maternal Genetic Effects		
	LS0_d_	LS3_d_	LS4_d_	LS0_m_	LS3_m_	LS4_m_
LS0_d_	0.139 * ± 0.049	0.939 * ± 0.071	0.924 * ± 0.080	−0.696 * ± 0.165	−0.897 * ± 0.105	−0.895 * ± 0.109
LS3_d_		0.170 * ± 0.036	0.999 * ± 0.001	−0.801 * ± 0.154	−0.830 * ± 0.109	−0.831 * ± 0.112
LS4_d_			0.162 * ± 0.036	−0.808 * ± 0.143	−0.816 * ± 0.110	−0.817 * ± 0.111
LS0_m_				0.216 * ± 0.051	0.862 * ± 0.095	0.867 * ± 0.095
LS3_m_					0.244 * ± 0.063	1.000 * ± 0.000
LS4_m_						0.238 * ± 0.061

* Significant at *p* < 0.05 by Z-test.

**Table 5 animals-12-03295-t005:** Direct and maternal heritability estimates (diagonals), genetic correlations (upper diagonals), and SE (lower diagonals) of piglet weight before weaning from a three-trait animal model with maternal genetic effects.

Trait	Direct Genetic			Maternal Genetic		
	MW0_d_	MW3_d_	MW4_d_	MW0_m_	MW3_m_	MW4_m_
MW0_d_	0.105 ± 0.066	0.926 * ± 0.184	0.874 * ± 0.260	−0.660 * ± 0.379	−0.682 * ± 0.221	−0.402 ± 0.298
MW3_d_		0.105 * ± 0.043	0.989 * ± 0.029	−0.500 ± 0.432	−0.431 ± 0.400	−0.145 ± 0.454
MW4_d_			0.095 * ± 0.043	−0.370 ± 0.533	−0.351 ± 0.463	−0.084 ± 0.481
MW0_m_				0.220 * ± 0.074	0.565 * ± 0.226	0.322 ± 0.261
MW3_m_					0.098 * ± 0.055	0.937 * ± 0.051
MW4_m_						0.118 * ± 0.053

* Significant at *p* < 0.05 by Z-test.

## Data Availability

Data are available only upon agreement with the breeding organization and should be requested directly from the authors.

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
