# Peer review of "Direct–Maternal Genetic Parameters for Litter Size and Body Weight of Piglets of a New Black Breed for the Taiwan Black Hog Market"

_animals, 2022, doi:10.3390/ani12233295_

Round 1
Reviewer 1 Report
General comment
The manuscript has rigorously investigated direct-maternal genetic parameters for litter size and body weight of piglet of a new black breed for Taiwan black hog market. They showed that maternal genetic effects were not negligible in their study, and thus multiple-trait animal model with maternal genetic effects and full pedigree was recommended to assist future pig breeding decisions in this new breed. However, several issues need to be solved before considering publication. I will present my points as follow:
Lines 96 to 97: were all individuals from a single farm? Or they were from multiple farms? If they were from different farms, a fixed effect of farm should be considered in the model.
Lines 119 to 122: 1) before finalizing the model, were the significances of fixed and random effects tested? If so, you may need to mention the significance of fixed and random effects for each studied trait in the manuscript. If not, involving the not significant fixed or random effect(s) in the final model may not be appropriate. 2) Just wondering if some individuals had repeated measurements for some studied traits? If so, the model may need to consider involving random permanent environmental and common-litter effects.
Lines 167 to 168: may consider adding the CV and the total number of records for each trait in Table 1.
Line 179: “….included in the model, significant and strong…..”, the significant level needs to be stated here, like P<0.05? or P<0.01?
Lines 181 to 183: “The strong positive genetic correlations between females’ littering traits from farrowing to weaning are due to the existence of pleiotropy in the traits evaluated as expected.” Several causes could lead to a strong/significant genetic correlation between two traits, such as linkage disequilibrium, biological pleiotropy, mediated pleiotropy, etc. The statement here is very confident, but the results from this manuscript cannot support this statement. May consider citing some references to support this (the studies/papers proved the existence of pleiotropy in those studied traits).
Lines 183 to 185: same comment as above.
Lines 187 to 189: this sentence is not very clear to me, may consider revising it.
Lines 190 to 193: Table 2, 1) I noticed all the genetic correlations between two traits are greater than their phenotypic correlations, but this point was not mentioned or discussed in the manuscript. May consider adding some related discussion; 2) It would be interesting to see the genetic and phenotypic correlations between litter size traits and body weight traits.
Lines 202 to 205: “Possible explanation might be due to more strong negative association between direct and maternal genetic effects within litter size traits (ranged from –0.814 to –0.757), and less strong association within piglet weight traits (ranged from –0.745 to –0.022).” may need some references to support this statement.
Lines 216 to 211: Table 3: the SEs were shown for the values under h2d, h2m, rdm columns but not for the values under h2r and h2T columns. May consider adding SEs for the values under h2r and h2T columns to keep the consistency of the table.
Lines 237 to 240: Table 4: The way of presenting the results in Table 4 is very confusing, should consider revising it. For example, removing the line between the LS4d row and LS0m row and having SEs right after the genetic correlations instead of separating them etc…
Lines 258 to 261: Table 5: same comments as comments for table 4.
Line 263: “ Descriptive Statistics”, the discussion here is not related to the content in “3.1. Descriptive Statistics” and what Table 1 showed.
Lines 280 to 282: references are needed to support this statement.
Lines 282 to 283: “ In this study, the h2 estimates reached moderate level for traits evaluated with range of 0.146–0.253….” h2 lower than 0.2 is usually treated as low level. May consider changing to “low-to-moderate”. In the meantime, several different h2 were estimated in this study, you may need to specify which one of them you mentioned here.
Line 294: “…which environmental effects could explain this situation.”, need to provide some details to explain how and what environmental effects influence this population(s).
Line 296:”… weak environmental correlation between these two…” the environmental correlations between traits were not estimated in this study; thus, “environmental correlation” cannot be used to explain the finding.
Lines 338 to 339 “ One possible explanation is that the trait is influenced by both direct and maternal genetic effects (i.e., h2d > 0 and h2m > 0), but these two effects oppose each other”. References are needed to support it.
Reviewer 2 Report
This study estimated genetic parameters of litter sizes and piglet weights of KHAPS black sow. The findings revealed the importance of maternal genetic effects, which significantly affects the selection accuracy of pig breeding. This study is well designed and generates interesting findings to future selective pig breeding. Some concerns need to be addressed.
1. In the discussion section, can the authors provide some explanations on how the maternal genetic effects affect the estimates of genetic correlations and heritability.
2. Conclusions in section 5 were a little of reductant, and it could be refined and improved.
Round 2
Reviewer 1 Report
The authors address my questions and comments. I just have one more comment. Please consider including a brief explanation of why random permanent environmental and common-litter effects were not involved in the models. I am sure that readers will have the same question why there were not involved. I would like to see what was written in the author's response somewhat reflected in the manuscript.
